# Microstructure and Fracture Toughness of Fe–Nb Dissimilar Welded Joints

Qiaoling Chu [1,2,3,*], Lin Zhang [1], Tuo Xia [1], Peng Cheng [4], Jianming Zheng [5], Min Zhang [1,*], Jihong Li [1], Fuxue Yan [1] and Cheng Yan [3,*]

[1]  School of Materials and Engineering, Xi'an University of Technology, Xi'an 710048, China; 2200121193@stu.xaut.edu.cn (L.Z.); 2200121103@stu.xaut.edu.cn (T.X.); lijihong@xaut.edu.cn (J.L.); yanfuxue@126.com (F.Y.)
[2]  CGN-DELTA (Jiangsu) Plastic & Chemical Co., Ltd., Taicang 215421, China
[3]  School of Mechanical, Medical and Process Engineering, Science and Engineering Faculty, Queensland University of Technology (QUT), Brisbane, QLD 4001, Australia
[4]  CGN Advanced Materials Technology (Suzhou) Co., Ltd., Taicang 215400, China; chengpeng@cgnam.cn
[5]  School of Mechanical and Precision Instrument Engineering, Xi'an University of Technology, Xi'an 710048, China; zjm@xaut.edu.cn
[*]  Correspondence: chuqiaoling@xaut.edu.cn (Q.C.); zhmmn@xaut.edu.cn (M.Z.); c2.yan@qut.edu.au (C.Y.)

**Abstract:** The relation between the microstructure and mechanical properties of the Fe–Nb dissimilar joint were investigated using nanoindentation. The weld metal consists mainly of $Fe_2Nb$, $\alpha$-Fe + $Fe_2Nb$, Nb (s,s) and $Fe_7Nb_6$ phases. Radial cracks initiate from the corners of the impressions on the $Fe_2Nb$ phase (~20.5 GPa) when subjected to a peak load of 300 mN, whereas the fine lamellar structures ($\alpha$-Fe + $Fe_2Nb$) with an average hardness of 6.5 GPa are free from cracks. The calculated fracture toughness of the $Fe_2Nb$ intermetallics is $1.41 \pm 0.53$ MPam$^{1/2}$. A simplified scenario of weld formation together with the thermal cycle is proposed to elaborate the way local phase determined the mechanical properties.

**Keywords:** fracture toughness; welding; intermetallics; nanoindentation

## 1. Introduction

The welding of dissimilar materials is always favorable for the industries because of its potential benefits on increasing design flexibility and reducing cost [1,2]. Niobium (Nb) is a refractory metal which has high thermal stability and excellent mechanical properties at elevated temperatures [3–5]. Nb acts in a wide variety of steels and Fe-containing alloys for ambient and high temperatures in multiple ways. Therefore, joining Nb with Fe has both technical and economic advantages. The formation of brittle and hard intermetallic compounds between Fe–Nb dissimilar joints is a very important issue. Budkin et al. [6] joined Nb with 12Cr8Ni10Ti steel via electron beam welding. It was reported that a continuous layer of $Fe_2Nb$ intermetallics at the interface deteriorated the mechanical properties of the weld due to high brittleness and hardness. Zhao et al. [7] used niobium as an intermediate layer in the hot-roll bonding of a Ti6Al4V alloy and Cr18Ni10Ti stainless steel. A layer of intermetallic compound between Nb and the weld zone was formed. Cracks were also found in the intermetallic layer. Kumar et al. [8] used the vacuum soldering method to join Nb and 316L. Although no brittle intermetallic layer formed, the joint had a low tensile strength. Compared to the brazing and solid-state welding, fusion welding is suitable for dissimilar materials and could obtain higher strength joints. Baghjari et al. [9] used Ni intermediate layer to join Nb and 410 stainless steel with laser welding. Fine cellular structure with Laves phase ($Fe_2Nb$) particles (~30–50 nm) was formed in the joint. Hajitabar et al. [10] investigated the weldability and mechanical properties of dissimilar joints between Nb–1Zr and 321 stainless steel produced by electron beam welding. The $Fe_7Nb_6$ and $Fe_2Nb$ intermetallic compounds formed in various regions

of the weld zone directly governed the mechanical properties of the weld. Therefore, it seems that the overall mechanical behavior of Fe–Nb dissimilar welds is highly controlled by the Fe–Nb intermetallics [11,12]. However, the fracture behavior, especially the fracture toughness of these Fe–Nb intermetallics, has not been well understood.

The nanoindentation technique which could evaluate the mechanical behavior at the nanoscale is increasingly used [13]. During the indentation on brittle materials, cracks may be induced by the impressions. Based on this, the facture toughness could be evaluated [14–16]. In this work, we attempted to study the cracking sensitivity of intermetallics in Fe–Nb dissimilar weld metal via nanoindentation tests. The phase formation mechanism in the resultant weld metal was explained using a simple model based on the phase diagrams and welding thermal cycles. This is a preliminary research for the further manufacturing of the Fe–Nb dissimilar welded structure.

## 2. Materials and Methods

A commercial low alloy steel plate (Q235, ~2 mm) was butt welded by pure Nb filler. The chemical compositions of materials used are listed in Table 1 [17]. The joints were single pass welded using Tungsten inert gas (TIG) method, where the current was 90 A and the voltage was 8 V. An SEM (JEOL-7001F, Japan Electron Optics Ltd., Tokyo, Japan) equipped with an EBSD detector (Oxford) was used to examine the microstructures in weld metal. X-ray diffraction (XRD) (Rigaku Corporation, Tokyo, Japan) patterns were collected on the cross-sections using a Rigaku-binary diffractometer (Rigaku SmartLab, Wilmington, MA, USA) with a Cu target. Hardness (H) and elastic modulus (E), were determined by Hysitron Triboindenter TI-950 with a Berkovich tip (Bruker Nano GmbH, Berlin, Germany). The Berkovich tip is a standard nanoindentation probe and is the best one for most bulk samples. A peak load of 8 mN with total 120 points were set. The measured hardness of each phase was averaged from at least ten indentations.

**Table 1.** Chemical compositions of materials used (wt.%).

| Materials | C | Si | Mn | Cu | Nb | Fe |
|---|---|---|---|---|---|---|
| Q235 (base metal) | 0.20 | 0.35 | 1.4 | - | - | Balance |
| Nb (filler) | 0.005 | 0.002 | - | 0.001 | Balance | 0.0014 |

MTS-G200 nanoindentation (MTS Systems Corporation, Eden Prairie, MN, USA) was applied to determine the fracture toughness ($K_C$) with a Berkovich tip. A peak load of 300 mN was applied. Figure 1 shows the crack dimensions measured from a Vickers's impression. In this work, $K_C$ was calculated by Equation (1) [16,18–21].

$$K_C = \chi_v \left(\frac{a}{l}\right)^{1/2} \left(\frac{E}{H}\right)^{2/3} \frac{P_{max}}{c^{3/2}} \tag{1}$$

where $P$max is the maximum indentation load, $a$ is the length from the center of the indentation to the corner, $l$ is the length of the crack from the indentation corner, $c$ is the average crack length from the center of the indentation to the crack tip ($c = l + a$), $E$ is the Young's modulus, $H$ is the material hardness, and $\chi_v$ is an empirical constant which depends on the indenter geometry. For a Berkovich indenter, $\chi_v$ is found to be 0.016 [16].

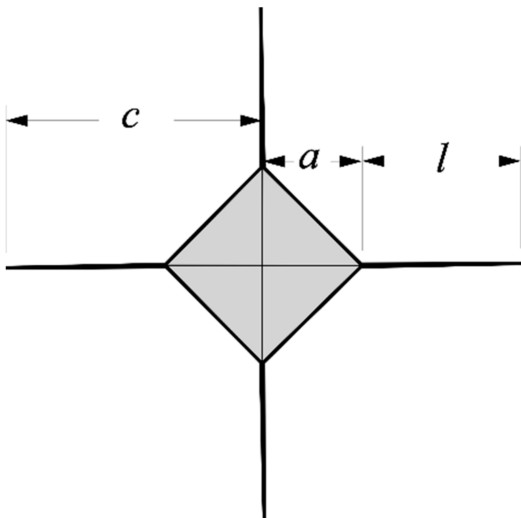

**Figure 1.** Illustration of surface crack geometry resulting from indentation with a Vickers pyramid.

## 3. Results and Discussion

The cross-section of the Fe–Nb welded joint is shown in Figure 2a. Severe cracks are observed in the Nb weld metal. The corresponding EDS line scanning results across the weld metal is plotted in Figure 2b, where the change in composition is drastic near the Fe–Nb weld metal (WM) interfaces and the weld metal presents alternative variation of Fe and Nb elements.

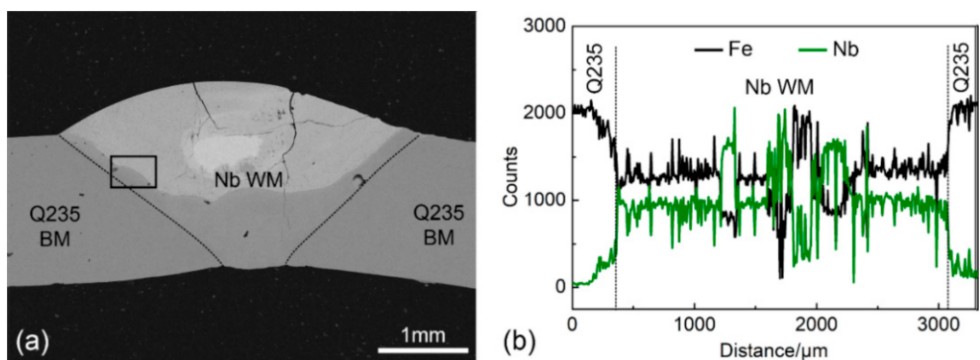

**Figure 2.** (**a**) The cross section of Fe–Nb welded joint; (**b**) EDS line scanning across the weld. BM: base metal; WM: weld metal.

Figure 3 shows the enlarged view of rectangle in Figure 2a. The compositions of typical phases determined by EDS analysis are listed in Table 2. The hardness obtained here was tested by the Berkovich tip with a peak load of 8 mN. The region adjacent to the Q235 base metal consists mainly of fine lamellar structure ($\alpha$-Fe + Fe$_2$Nb), as shown in Figure 3a,b. These lamellar structures are the eutectic products with an average hardness of 6.5 GPa. Some Nb solid solution with extremely high hardness value (47.3 GPa) is dispersed in these lamellar structures, as shown in Figure 3c. This characteristic distribution is confirmed by the corresponding EBSD phase mapping in Figure 3e. It should be noted that due to the similar lattice parameters between the $\alpha$-Fe and Nb phases, these observed Nb dendrites could not be recognized in the EBSD phase maps. Cellular structures are formed adjacent to the lamellar structure ($\alpha$-Fe + Fe$_2$Nb), as shown in Figure 3d. These cellular structures primarily consist of Fe$_2$Nb intermetallics and have an average hardness of 20.5 GPa, as confirmed by the corresponding EBSD phase mapping in Figure 3f. Figure 3g presents the microstructure in the center of Nb weld metal. Combined with the Fe–Nb binary

diagram and EDS data in Table 2, this microstructure present here is fully eutectic products ($Fe_7Nb_6$ + Nb) with an average hardness of 13.1 GPa. A similar microstructure was also reported by Voß et al. [12]. In this work, the cracks observed mainly initiate and propagate in $Fe_2Nb$ intermetallics, suggesting the brittle nature of this phase.

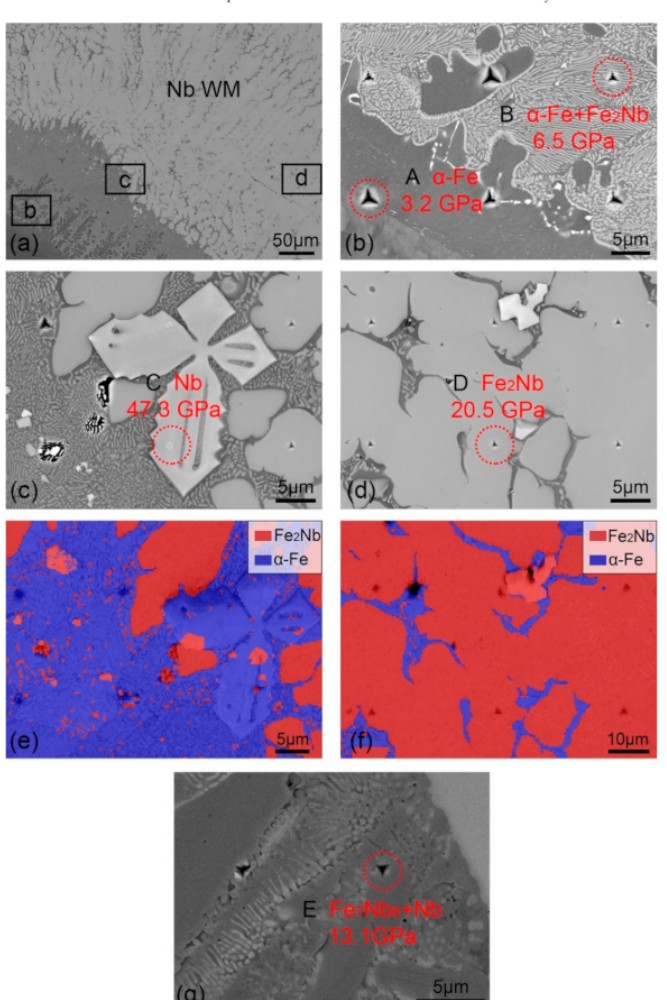

**Figure 3.** Microstructure in the Fe–Nb weld metal: (**a**) low magnification image at the Q235 BM/Nb WM interface; (**b**–**d**) the enlarged views of rectangles in (**a**); (**e**,**f**) the corresponding EBSD phase maps (superimposed with band contrast images) of (**c**,**d**), respectively; (**g**) the microstructure in the center of Nb weld metal.

**Table 2.** Chemical compositions of typical phases in Nb weld metal (at. %).

| Regions | Fe | Nb | Possible Phases | Hardness/GPa | Elastic Modulus/GPa |
|---|---|---|---|---|---|
| A | 98.60 | 1.40 | α-Fe | 3.2 | 206.5 |
| B | 89.68 | 10.32 | α-Fe + $Fe_2Nb$ | 6.5 | 206.5 |
| C | 10.12 | 89.88 | Nb | 47.3 | 378.9 |
| D | 69.55 | 30.45 | $Fe_2Nb$ | 20.5 | 275.3 |
| E | 45.36 | 54.64 | $Fe_7Nb_6$ + Nb | 13.1 | 176.1 |

The XRD patterns in Figure 4 confirm the above phase constituents. The weld metal is dominated by $Fe_2Nb$ intermetallics. The regions near the Q235 base metal are rich in α-Fe + $Fe_2Nb$ compounds and the center weld consists mainly of $Fe_7Nb_6$ + Nb compounds.

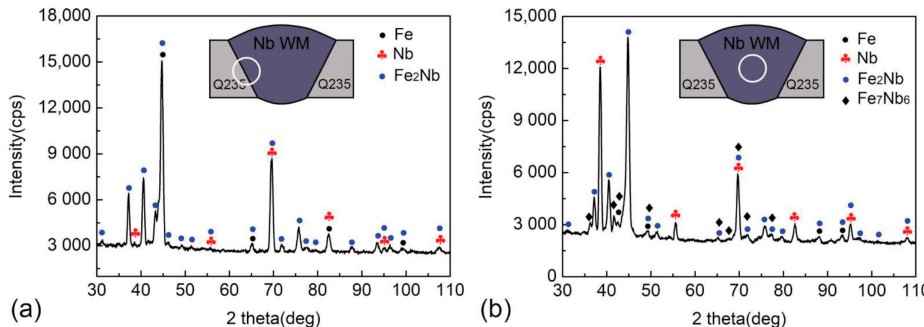

**Figure 4.** XRD patterns: (**a**) At Q235 BM/Nb WM interface; and (**b**) in the center of Nb weld metal.

Figure 5a shows the global hardness distribution. The $Fe_2Nb$ intermetallics which is dominant in the weld metal show the higher hardness than $\alpha$-Fe + $Fe_2Nb$ and $Fe_7Nb_6$ + Nb compounds. The load–displacement (P–h) curves of typical phases ($\alpha$-Fe, $\alpha$-Fe + $Fe_2Nb$, $Fe_2Nb$ and $Fe_7Nb_6$ + Nb) are shown in Figure 5b. The P–h curve of $Fe_2Nb$ exhibits some serrations (indicated by black arrows) during loading, whereas that of $\alpha$-Fe + $Fe_2Nb$ and $\alpha$-Fe are quite smooth. These serrations signal the onset of plasticity in the indented materials [22–24]. The corresponding $Fe_2Nb$ impression (Figure 5d) reveals some pile-up (indicated by black arrows), whereas the $\alpha$-Fe + $Fe_2Nb$ impression (Figure 5c) shows no obvious plasticity morphology. The maximal indenter displacement ($h_{max}$) upon the 8 mN peak load is much larger for $\alpha$-Fe (~355 nm) and $\alpha$-Fe + $Fe_2Nb$ compounds (~249 nm) compared to that of $Fe_2Nb$ phase (~151 nm) and $Fe_7Nb_6$ + Nb (~195 nm).

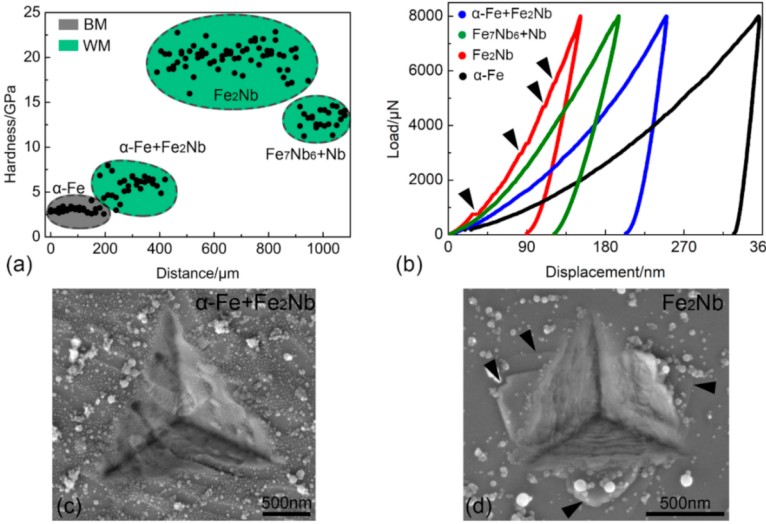

**Figure 5.** (**a**) Hardness distribution in the Nb welded joint; (**b**) P–h curves of typical phases; (**c,d**) $\alpha$-Fe + $Fe_2Nb$ and $Fe_2Nb$ impressions, respectively.

The representative impressions with a peak load of 300 mN are shown in Figure 6. $\alpha$-Fe + $Fe_2Nb$ compounds show no obvious cracks under 300 mN, as shown in Figure 6a. Figure 6b–d are the impressions of $Fe_2Nb$ phase, where radial cracks are observed. The cracks initiate from one or two impression corners and propagate along a straight-line path. Pile-ups are also observed around these impression edges. The average crack length (c) is systematically recorded for every impression. In this study, only indentations with well developed cracks and without chipping are applied to calculate the $K_C$ values. According to Equation (1), a mean value and standard deviation of $1.41 \pm 0.53$ MPam$^{1/2}$ is obtained for $Fe_2Nb$ intermetallics. Such lower value also confirms the brittle nature of $Fe_2Nb$ intermetallics.

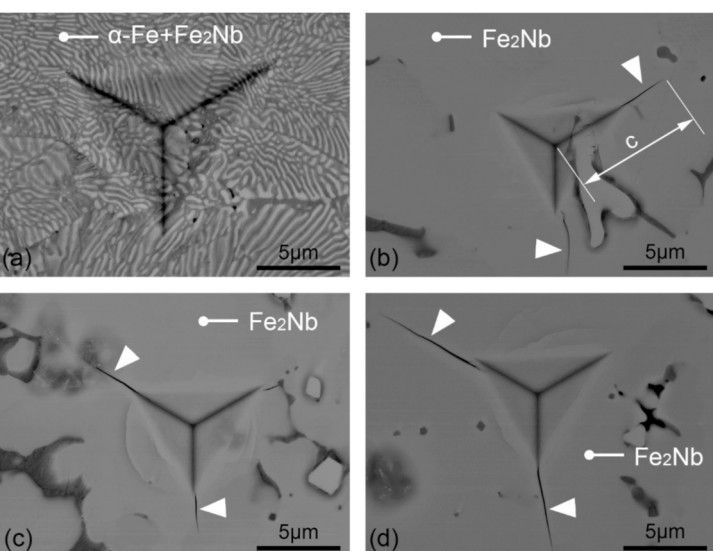

**Figure 6.** Backscattered electron (BSE) images of impressions in (**a**) the α-Fe + Fe$_2$Nb region and (**b**–**d**) the Fe$_2$Nb regions.

It is well known that the microstructure transforms in the weld metal varies over time during the fusion welding. The macroscopic grain structure in gas–tungsten arc welding is controlled by a combination of the thermal conditions that prevail at the solid–liquid interface and the crystal growth rate [25]. In order to further develop the correlation between the microstructure and mechanical properties in the weld metal, we break down the whole transformations into a number of stages, along with thermal cycles and an Fe–Nb phase diagram [11], shown in Figure 7.

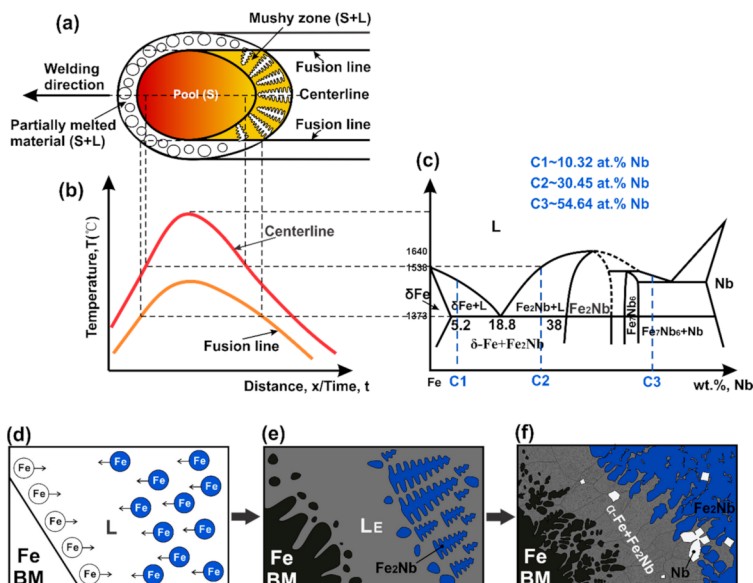

**Figure 7.** Schematic diagrams showing the progression of the phase transformations in Nb weld metals during the solidification: (**a**) microstructure of the solid plus liquid around the weld pool; (**b**) thermal cycles; (**c**) part of Fe–Nb binary phase diagram; (**d**–**f**) microstructure transformations in the Fe-rich portion. L: liquid; S: solid.

According to the microstructure in Figure 3, the Nb element exhibits heterogeneous distribution. We classify the Fe–Nb welded metal as two parts, the Fe-rich part (indicated as C1 and C2) and the Nb-rich part (indicated as C3), shown in Figure 7c. Figure 7d–f show

the microstructure evolution in the Fe-rich part. Generally, the solidification of the melt initiates from the base metal for the pure metal. However, there perhaps two reactions initiate in the weld pool simultaneously due to the high solidification temperature of the $Fe_2Nb$ phase (1641 °C). Cellular dendrites $Fe_2Nb$ form in the liquid with the concentration of C2 (L ↔ $L_E$ + $Fe_2Nb$), while dendrites δ-Fe form near the Q235 boundary with the concentration of C1 (L ↔ $L_E$ + δ-Fe) [11,12], as shown in Figure 7e. The region with such characteristics is called the mushy zone [26]. As shown in Figure 7a, this mushy zone behind the trailing portion of the pool boundary consists of solid dendrites (S) and the interdendritic liquid (L). The concentration of the residual liquid L in this region is close to $L_E$. When its temperature is below 1373 °C, an eutectic reaction occurs ($L_E$ ↔ δ-Fe + $Fe_2Nb$) and produces the typical lamellar structure ($Fe_2Nb$) dispersed in the δ-Fe matrix, as shown in Figures 3b and 7f. The residual liquid is then gradually depleted and hereafter some solid-state transformations form, such as δ-Fe ↔ γ-Fe + $Fe_2Nb$ at the temperature of 1183 °C and γ-Fe + $Fe_2Nb$ ↔ α-Fe at the temperature of 943 °C [11,12]. Due to the high cooling rate and convection effect in the TIG pool, some Nb drops are involved in the Fe-rich part and form the Nb dendrites in Figure 7f. Approaching the center weld, the Nb concentration increases. At this part the Nb concentration is around 54.64 at.% (listed in Table 2), indicated as C3. The microstructure in this region is nearly fully eutectic products (L ↔ $Fe_7Nb_6$ + Nb, ~1508 °C).

Nb is an essential alloying element for steels, which makes the Fe–Nb system of high interest. Nb acts in a wide variety of steels and Fe-containing alloys for ambient and high temperatures in multiple ways, e.g., through solid solution, micro-carbide formation, and the formation of intermetallic phases [12]. Particularly the latter aspect recently attracts increasing attention as alloys that are strengthened by fine precipitates of the Laves phase $Fe_2Nb$ offer considerable strength and creep resistance also at elevated temperatures [27–29]. The analysis of this study confirms only two intermetallic phases existing in Fe–Nb system, the hexagonal C14-type $Fe_2Nb$ Laves phase and the μ phase $Fe_7Nb_6$. The Laves phase $Fe_2Nb$ extending to the Fe-rich side and μ phase $Fe_7Nb_6$ to the Nb-rich side. Together with the nanoindentation results, the Laves phase $Fe_2Nb$ is sensitive to cracking. The eutectic products in the Fe-rich part (α-Fe + $Fe_2Nb$) and Nb-rich part ($Fe_7Nb_6$ + Nb) show relative high cracking resistance. This paper determines the basic mechanical behavior of Laves phases $Fe_2Nb$ in an Fe–Nb dissimilar joint. This is a preliminary research for further manufacturing of Fe–Nb dissimilar welded structure. As the direct joining of Nb to steel will encountered severe problems because of differences in their physical and chemical properties and formation of very brittle Fe–Nb intermetallics (especially the $Fe_2Nb$ intermetallics) in the weld. The mechanical strength of such welds can be enhanced by applying suitable interlayer that modifies final phase composition, which our future study will be focused on.

## 4. Conclusions

In this paper, the correlation between the microstructure and mechanical properties in Fe–Nb weld metal was developed by nanoindentation. Several points about the phase selection and solidification sequence in the weld metals were suggested by combining crystallographic and morphological information. The resultant weld metal was primarily composed of two parts, the Fe-rich part which consisted of $Fe_2Nb$ and α-Fe + $Fe_2Nb$ compounds and the Nb-rich part which consisted of $Fe_7Nb_6$ + Nb compounds. The fracture toughness of the typical phases was evaluated by nanoindentation with a peak load of 300 mN. For the $Fe_2Nb$ intermetallic phase, the estimated average hardness and the fracture toughness were 20.5 GPa and 1.41 ± 0.53 $MPam^{1/2}$, respectively. The eutectic compounds (α-Fe + $Fe_2Nb$) are free of cracks, showing an average hardness of 6.5 GPa.

**Author Contributions:** Conceptualization, Q.C. and M.Z.; methodology, J.L.; validation, F.Y., L.Z. and T.X.; investigation, Q.C.; resources, P.C.; writing—original draft preparation, Q.C.; writing—review and editing, Q.C. and C.Y.; supervision, J.Z.; project administration, M.Z.; funding acquisition, Q.C., M.Z. and C.Y. All authors have read and agreed to the published version of the manuscript.

**Funding:** This research was funded by National Natural Science Foundation of China, grant number 51904243, Natural Science Foundation of Shaanxi Provincial Department of Education, grant number 19JK0573, Natural Science Foundation of Shaanxi Province, grant number 2019JQ-284 and 2019JZ-31, China Postdoctoral Science Foundation, grant number 2019M653704 and the Australian Research Council Discovery Project, grant number DP180102003.

**Institutional Review Board Statement:** Not applicable.

**Informed Consent Statement:** Informed consent was obtained from all subjects involved in the study.

**Data Availability Statement:** Data sharing not applicable. The raw data required to reproduce these findings cannot be shared at this time as the data also forms part of an ongoing study.

**Conflicts of Interest:** The authors declare no conflict of interest.

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
