# Peer review of "Microstructure and Fracture Toughness of Fe–Nb Dissimilar Welded Joints"

_metals, doi:10.3390/met11010086_

Round 1

Reviewer 1 Report

This paper studies the fracture toughness of Fe-Nb of dissimilar material welds. The authors attempt to correlate the microstructure and mechanical properties using nano indentation tests.  The authors also report on observed phases in the weld.

The paper reads well and clear, however please check the following comments especially on literature review before the manuscript can be accepted for publication:

The literature review is almost non existent, only one reference was cited and discussed, the authors must carry out proper literature review on past studies on similar welds or dissimilar welds in general and report what was done and what were the main findings, and they show how their work differs from past studies.

Table 1 is missing references, where did this data was taken from?

Equation 1 needes to be referenced too.

Line 121, can the authors explain why there s no signs of plasticity?

The impression loads in Fig5 are a bit not clear, have the author tried applying higher loads for example 500 or more to see if the indent becomes more clear on the surface?

Reviewer 2 Report

The manuscript "Microstructure and fracture toughness of Fe-Nb dissimilar welded joints" has been reviewed. It deals with the fracture toughness of intermetallics investigated by nanoindentantation technique. 

Line 51: TIG process parameters are not detailed. Please add!

Line 56: please specify X-ray source.

Line 56: The choice of the Berkovich tip is not explained. Please add!

Line 62 - Fig. 1: It is not clear if Berkovich tip is employed (line 56) or Vickers indenter (line 62 - Fig. 1). Please clarify!

Equation 1: Xv is indicated instead of Xd.

Table 1: the adoption of nearly pure Nb as filler is a partiicularly worsening condition for the manufacturing of the weld. How is it motivated? This is not a real welding condition!

Table 2: Hardness: please specify the indenter, method and load used.

References: Journal title is not in compliance with Metals standard.

Round 2

Reviewer 1 Report

All questions have been answered, the paper can be accepted for publication

Reviewer 2 Report

accept in present form